# Isovolumetric Contraction as a Marker of Ventricular Performance in Patients with Afterload Mismatch

**DOI:** 10.3390/diagnostics13081366

**Published:** 2023-04-07

**Authors:** Theodoros Sinanis, Eleftherios Markidis, Symeon Evangelos Mavroudeas, Emmanouil Sideras, Evangelos Vittorakis, Eftychios Vittorakis

**Affiliations:** 1Cardiology Department and Cardiology Intensive Care Unit, “Agios Georgios” General Hospital of Chania, p.c. 73100 Chania, Greecevittorakis.vagelis@gmail.com (E.V.); 2Department of Microbiology, “Iuliu-Hatieganu” University of Medicine and Pharmacy Cluj-Napoca, p.c. 400349 Cluj-Napoca, Romania

**Keywords:** end-systolic elastance, pulmonary arterial hypertension, severe aortic stenosis, Frank–Starling mechanism, isovolumetric contraction, myocardial contractility, MADIT study, SCD-HeFT study, Fick method

## Abstract

Introduction: The evaluation of myocardial contractility is essential in cardiology practice. The gold standard for this evaluation is the end-systolic elastance, but it the method involved is complex. Echocardiographic measurement of the ejection fraction (EF) is the most commonly used parameter in clinical practice, but it has significant limitations, especially in patients with afterload mismatch. In this study, the area under the curve (AUC) of the isovolumetric contraction was measured to evaluate the myocardial contractility in patients with pulmonary arterial hypertension and severe aortic stenosis. Methods: 110 patients with severe aortic stenosis and pulmonary arterial hypertension were included in this study. The AUC of the isovolumetric contraction was measured using pressure curves of the right ventricle–pulmonary artery and left ventricle–aorta ascendens. This AUC was then correlated with the echocardiographically measured EF, stroke volume (SV), and total ventricular work. Results: The AUC of the isovolumetric contraction showed a statistically significant correlation with the EF of the corresponding ventricle (*p* < 0.0001). Both the AUC of the isovolumetric contraction and the EF showed a statistically significant correlation with the total work of the ventricle (AUC: R2 0.49, *p* < 0.001; EF: R2 0.51, *p* < 0.001). However, the SV only showed a statistically significant correlation with the EF. A statistically significant one-sample t-test could be found for the EF (decreased, *p* < 0.001) and for the AUC of the isovolumetric contraction (increased, *p* < 0.001), but not for the total work of the ventricle. Conclusion: The AUC space of the isovolumetric contraction is a useful marker of ventricular performance in patients with afterload mismatch, showing a statistically significant correlation with the EF and the total ventricular work. This method may have potential for use in clinical practice, especially in challenging cardiological cases. However, further studies are needed to evaluate its usefulness in healthy individuals and in other clinical scenarios.

## 1. Introduction

The current gold standard method for the non-invasive evaluation of ventricular function is the ejection fraction [1]. The fact that left ventricular ejection fraction (LVEF) is used as the primary inclusion criteria in MADIT 2 [2] and in ScD HEFT [3], both pivotal ICD treatment studies, makes its value undisputed. As a result, the current heart failure guidelines of the European Society of Cardiology [1] also categorize patients according to LVEF. Despite its major role in current practice, LVEF has major limitations when evaluating ventricular function and performance, mainly because it ignores the spatial deformation of the myocardium. For this reason, studies have shown that global longitudinal strain (GLS) is a superior predictor of outcome and a better assessment tool for global left ventricular systolic function [4,5]. Nevertheless, the use of strain rate and longitudinal strain in the current guidelines is not without limitations and pertains only to specific situations [1]. Although the dependency of LVEF on preload is known (according to the Frank–Starling mechanism), its limitations are more significant in populations with afterload mismatch, especially when this mismatch is severe. This is why cardiac catheterization with the measurement of ventricular pressure at all times is mandatory when diagnosing some patients with severe aortic stenosis and mixed echocardiographic findings [6], and all patients with pulmonary arterial hypertension [6]. On the other hand, the most accurate way to evaluate the performance of a piston pump is to measure its instantaneous calculation of total product, defined by both pressure and generated flow, which is very similar to the Frank–Starling curve used in cardiac pathophysiology [7]. Since these calculations may be different at any given moment, head pressure, which is the result of the pressure produced by the pump when the flow is zero, is often used [8,9]. The phase of systole when both valves are closed (and, as a result, the productive flow is zero) is the isovolumetric contraction. The aim of this study is to investigate the clinical and pathophysiological significance of isovolumetric contraction and its relation to ventricular performance in patients with afterload mismatch—specifically, patients with severe aortic valve stenosis and patients with pulmonary arterial hypertension.

## 2. Methods

In our study, we collected patients admitted to the cardiology department of the General Hospital of Chania, Crete, Greece. All of the included patients were over 18 years of age and were undergoing right and left cardiac catheterization for diagnostic purposes. Only patients with confirmed severe aortic stenosis or pulmonary arterial hypertension were investigated further, while patients with other diagnoses, mild or moderate aortic stenosis, or pulmonary venous hypertension were excluded. The ejection fraction was measured echocardiographically using the Simpson’s biplane method, which involves using the end-diastolic and end-systolic apical 4- and 2-chamber views to estimate the left ventricle volume and calculate the ejection fraction. All echocardiograms were performed by the same operator during the same hospital stay in which cardiac catheterization was performed.

The stroke volume was calculated using the Fick method, which is based on the principle that oxygen consumption is equal to the product of the organ’s blood flow (CO) and the difference in the concentration of the substance in the arterial and venous circulation (arterial–venous oxygen difference) [6]. In the patients with aortic stenosis, pressure measurements were performed in the left ventricle, while in the patients with pulmonary artery hypertension, they were performed in the ascending aorta, the right ventricle, and the pulmonary artery. We used the isovolumetric contraction phase, which occurs during early systole when the ventricles contract without any corresponding volume change (isometrically), to calculate the duration and pressure of the isovolumetric contraction. The pressure curves of the left ventricle with the ascending aorta and the right ventricle with the pulmonary artery were used. The time of the isovolumetric contraction was determined by calculating the interval between the R wave of the electrocardiogram and the beginning of the rise in pressure in the ascending aorta or the pulmonary artery (the time point at which the aortic valve and the pulmonary valve open). This time interval was then transferred to the corresponding ventricular pressure curve, starting at the point of the end-diastolic pressure. The area under the curve was then calculated, and it was assumed that the area was an orthogonal triangle. 

The product of cardiac output and the maximum pressure difference (which is the difference between the maximum ventricular pressure and the end-diastolic ventricular pressure) is commonly defined as the total work. In this study, we measured three beats in patients with sinus rhythm and five beats in patients with atrial fibrillation, and the average value was used. The measurements were taken during normal breathing, and deep inspirations and expirations were excluded from the analysis.

## 3. Statistical Analysis

The evaluation was conducted using the SPSS (Statistical Package for the Social Sciences) 23.0G software for Windows. Descriptive statistics were used to determine the usual values, including arithmetic mean, median, standard deviation, standard error, and minimum and maximum values. The correlation coefficient r (Pearson correlation coefficient) and regression coefficient beta were used to describe the correlation between continuous variables. Multiple linear regression analyses were used to represent correlations with several other parameters and to test whether these correlations were significant. The study considered statistical significance to be R² > 0.5 in combination with *p* < 0.05. After the bivariate analysis, a multivariate analysis was carried out to test the independent influence of each parameter. Here, an error probability below 5% (*p* < 0.05) was also considered significant. Finally, a one-sample t-test was performed for each of these parameters.

## 4. Results

We developed our research protocol and obtained ethical approval from the ethical committee of the General Hospital of Chania to undertake this study with patients who met the research criteria. We evaluated a total of 110 patients who met the research criteria, out of whom 50 had severe aortic stenosis and 60 had pulmonary arterial hypertension. The exclusion criteria included patients with a bicuspid aortic valve and those with cardiomyopathies, multiple valvulopathies, or other anatomomorphological manifestations.

For the patients with severe aortic stenosis, native valve and concomitant mitral regurgitation were considered, and right and left cardiac catheterization were performed to evaluate the stenosis before a transfemoral aortic valve implantation (TAVI). For the patients with pulmonary arterial hypertension, tricuspid valve regurgitations of various gradations were observed, and right heart catheterization was performed for the diagnosis of pulmonary arterial hypertension only, and not as a follow-up. The patients were all untreated at the time of the right heart catheterization, and a medical treatment with a specific pulmonary arterial hypertension medication was initiated for all of the 60 subjects within 30 days post-diagnosis.

The patients were required to undergo angiography in the laboratory for catheterization to determine their endovascular pressures and receive a complete anatomic and functional assessment via echocardiography.

The entire patient cohort had a mean ejection fraction of 41% (49% for the aortic stenosis subgroup and 35% for the pulmonary arterial hypertension subgroup). The corresponding stroke volumes were 63 mL (total), 56 mL (aortic stenosis subgroup), and 68 mL (pulmonary arterial hypertension subgroup). The end-diastolic pressure for the entire patient cohort was 14 mmHg (21 mmHg for the aortic stenosis subgroup and 9 mmHg for the pulmonary arterial hypertension subgroup). The average maximal pressure generated by the left ventricle during systole was 121 mmHg for all patients, 199 mmHg for the aortic stenosis subgroup, and 68 mmHg for the pulmonary arterial hypertension subgroup. The time of isovolumetric contraction was 0.11 s for the entire cohort, 0.09 s for the aortic stenosis subgroup, and 0.12 s for the pulmonary arterial hypertension subgroup.

The AUC of the isovolumetric contraction was found to have a statistically significant correlation with the ejection fraction of the corresponding ventricle (*p*-value < 0.001).

Both the AUC of the isovolumetric contraction and the ejection fraction were found to be statistically significant with respect to the total work of the ventricle (the AUC had an R² of 0.49 and a *p*-value of 0.0001; the EF had an R² of 0.51 and a *p*-value of 0.0001). This significance remained even after performing a multivariate analysis for both the AUC and the ejection fraction.

A further correlation was tested between the ejection fraction and the AUC of the isovolumetric contraction with respect to the total work of the ventricle. A statistically significant correlation was found only for the ejection fraction, but not for the AUC of the ventricle.

One-sample t-tests were performed for the ejection fraction, the AUC of the isovolumetric contraction, and total work of the ventricle. The ejection fraction showed statistical significance (decreased, *p*-value 0.001), as did the AUC of the isovolumetric contraction (increased, *p*-value 0.001), but the total work of the ventricle did not.

## 5. Discussion

In this study, we investigated the pathophysiological and clinical significance of isovolumetric contraction in patients with diseases causing afterload mismatch in both the right and left ventricles. The pressure curve of the ventricle during diagnostic cardiac catheterization was used to analyze this. The conditions causing afterload overload were aortic valve stenosis and pulmonary arterial hypertension. We found a statistically significant correlation between both the AUC of the isovolumetric contraction and the echocardiographically measured ejection fraction and the total work of the ventricle, defined as the product of pressure and volume per systole. The ejection fraction was low in both subgroups, and the time of isovolumetric contraction was significantly prolonged. However, the total work of the ventricle did not differ significantly.

As was expected, the patients with aortic stenosis had much higher pressure compared with the patients with pulmonary arterial hypertension. Not only does the left ventricle produce greater pressure than the right, but when an afterload mismatch occurs, it is much higher in the left ventricle than in the right [10,11]. The time of isovolumetric contraction was prolonged in all cases, leading to a less steep rise in the ventricular pressure curve observed in the aortic stenosis and pulmonary arterial hypertension subgroups [12,13].

The ejection fraction was mildly reduced in the patients with aortic stenosis and moderately reduced in the patients with pulmonary arterial hypertension. The more reduced ejection fraction in the patients with pulmonary arterial hypertension compared with the patients with aortic stenosis corresponds to findings from international registries [14,15,16].

Each ventricle produces pressure and flow, and both the documented product of the maximum pressure and the stroke volume were used as the total work of the ventricle produced per heartbeat. Both the ejection fraction and the AUC space of the isovolumetric contraction showed a statistically significant correlation with this product. Additionally, the echocardiographically estimated ejection fraction and the AUC space of the isovolumetric contraction demonstrated a statistically significant correlation with each other.

Another significant finding of this study is that only the ejection fraction, and not the AUC space of the isovolumetric contraction, correlated with the stroke volume. The stroke volume in this statistical analysis was calculated using the Fick method during the right and left cardiac catheterization, whereas the ejection fraction was measured echocardiographically. Since the ejection fraction is defined as the stroke volume divided by the end-diastolic volume, the significance of this correlation is not surprising. It should be noted that the isovolumetric contraction shows no correlation with the stroke volume, most likely because they represent different products of the ventricle. The isovolumetric contraction occurs precisely at the point at which the flow is zero.

Hydraulic models of the cardiovascular system typically depict the ventricle as hydraulic piston pumps, which means that using only the ejection fraction to evaluate ventricular performance may not fully capture its function and could lead to diagnostic challenges. Most hydraulic models focus on the head pressure of the pump and the total work produced rather than the displacement of the piston [17,18,19,20,21,22]. In real-world settings, it is not feasible to monitor pressure and volume instantaneously. Thus, isovolumetric contraction, which occurs when both valves of each ventricle are closed, can provide an estimate of the head pressure produced in the absence of flow. In this study, the area under the curve of the isovolumetric contraction was used as a more accurate equivalent to evaluate ventricular performance during this phase of the cardiac cycle [21].

Compared with normal subjects, patients with afterload mismatch, regardless of etiology, exhibit a reduction in ejection fraction and an increase in the area under the curve of the isovolumetric contraction. However, there is no statistically significant change in the total work produced from each ventricle. Both changes can be a result of homometric and heterometric mechanisms of autoregulation in response to the increased afterload [22,23,24]. Both mechanisms are likely to influence both parameters, but it is more likely that isovolumetric contraction, being volume-independent by definition, represents an increase in myocardial contractility more precisely.

This is a retrospective monocentric study based on patients with an increase in afterload, specifically those with aortic stenosis and pulmonary hypertension. The presence of tricuspid regurgitation and/or mitral regurgitation was not considered, and these factors may be significant when studying isovolumetric contraction. A direct comparison with healthy adults was not performed, but known normal values were used for comparison. Comorbidities such as chronic renal failure and diabetes, which may increase systemic resistance in circulation, were not taken into account. The stroke volume of each ventricle was not directly measured, but was calculated using the Fick method and the Bergstra equation for the estimation of oxygen uptake [25].

It is necessary to mention certain limitations of this study. Some of our patients did not wish to participate in our study as research subjects, and our sample size was smaller than we would have liked due to the fact that our hospital caters to a small population. Our study only considered patients with pulmonary artery hypertension and aortic stenosis, and, therefore, it will be important to conduct further research with different scenarios and diagnostic tools to obtain a broader understanding of this isovolumetric marker. Additionally, the lack of experimental animals and research centers in our region hindered our ability to analyze the theoretical background in experimental animals. The COVID-19 pandemic has also been a major obstacle to our work as it has been difficult to work with our patients due to various healthcare protection reasons, and the population has been volatile and in need of protection.

We would like to emphasize the importance of this theoretical background knowledge and the need for future experimental studies to evaluate the statistical significance of this study and its importance for the management of patients’ treatment. Our future proposal is to evaluate isovolumetric contraction as a marker of ventricular performance in patients with afterload mismatch using experimental studies. For this experimental study, it will be important to receive assistance from experimental centers abroad and a establish a multi-research team from different research centers. It is important to note that the exit from the COVID-19 era is absolutely necessary as travel and communication bans have halted researchers’ communication and the exchange of information.

### Future Perspectives

Having established a statistically significant correlation with the ejection fraction in the clinical scenario of increased afterload, the next step will be to demonstrate a statistically significant correlation in every other possible clinical scenario, as well as in healthy individuals. However, the most accurate method to establish a correlation between the AUC during isovolumetric contraction and ventricular contractility is to correlate it with the end-systolic elastance measured using pressure–volume loops in an experimental environment. Only after establishing a correlation between the work produced during the isovolumetric contraction and the gold standard method for evaluating myocardial contractility, as well as systemic vascular resistance, can a further investigation be conducted on correlations in preload and/or afterload increases.

If all of the above hypotheses are proven, the potential perspectives are numerous. Using a simple pigtail catheter during cardiac catheterization to accurately measure myocardial contractility could provide a solution in almost every challenging cardiological case, including low-flow, low-gradient aortic stenosis (paradoxical or not) and multivalvular disease, or in determining whether a TAVI will effectively reduce concomitant mitral regurgitation. This method could also provide further information during high-risk PCI to identify patients in need of hemodynamic support (such as an Impella) based not only on statistically proven scores but also on individualized measurements.

A clinical prospective study involving healthy individuals or patients at risk of hemodynamic collapse may be unnecessary and ethically questionable. A more realistic approach for a prospective clinical study could involve patients undergoing a coronary angiogram prior to cardiac surgery for aortic or mitral valve disease, regardless of the treatment method (percutaneous or surgical).

## Data Availability

Not applicable.

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
