# Peer review of "Isovolumetric Contraction as a Marker of Ventricular Performance in Patients with Afterload Mismatch"

_diagnostics, 2023, doi:10.3390/diagnostics13081366_

Round 1

Reviewer 1 Report

The manuscript is well written but introduction has many details and matters not related exactly to your study regarding ICD  and so on 

it would be better to shorten that explanations 

Author Response

I would like to apologize to the reviewers for the delay in reviewing this manuscript. This was due to the fact that my house was burglarized and I had to deal with many issues.

I have asked an English native-speaking reviewer to check the manuscript for grammar and vocabulary mistakes.

Furthermore, I have summarized the introduction part in a more concise and clear manner, eliminating unnecessary information.

Thank you for your review report.

Reviewer 2 Report

Sinanis et al. aimed to explore if the isovolumetric contraction can be used as a marker of ventricular performance in patients with afterload mismatch. The results showed that the AUC of the isovolumetric contraction correlated significantly with ejection fraction and the total ventricular work, but not with the stroke volume.

There are major issues to address:

1.     I identified plagiarism in the text, which is not acceptable (for example “The Frank-Starling mechanism is a physiological principle that explains how the heart responds to changes in venous return. Increases in venous return cause the heart's chambers to fill with more blood, which then causes the heart to stretch and contract more forcefully, and pump more blood out to the rest of the bodyhttps://www.osmosis.org/learn/Frank-Starling_relationship#:~:text=The%20Frank%2DStarling%20mechanism%20is,the%20rest%20of%20the%20body.)

 2.     The figures don’t have any legend/title. Do they belong entirely to the authors or are they modified with permission of other authors?  

3.      The authors should revise the English language since it is almost incomprehensible. (raw 113 ‘ all our patients were above 18 years old undergoing a right a left cardiac catheterization for diagnostic purposes were included’, raw 123 ‘Fick’s method in which is based on…’, ‘in our study we collected patient who admitted’). The time of the verb should be kept the same (‘was calculated’, ‘we calculate’),)

4.     In the study I found confusions such as ‘heart failure’ is a cardiomyopathy etc. This is not acceptable. I am sorry, but I cannot follow the article, please re-write it correctly.

5.     What is the relevance of that detailed description of the MADIT trials in the introduction? The authors should keep that detailed theoretical information short and focus more on the topic of the study.

Author Response

I would like to apologize to the reviewers for the delay in reviewing this manuscript. The delay was caused by a burglary in my house, which required me to deal with numerous issues.

I have asked a native English-speaking reviewer to check the manuscript for any grammar and vocabulary mistakes.

Furthermore, I have summarized the introduction in a more concise and clear manner, removing unnecessary information.

I have addressed the plagiarism issue mentioned by the reviewer by revising the introduction as explained earlier.

I have also removed the figures from the article since they do not provide further explanation and could be difficult to assess as bibliographic material.

We have made efforts to resolve the issues and misunderstandings regarding cardiomyopathies and heart failure, which were identified by the reviewer.

Thank you for your review report.

Round 2

Reviewer 2 Report

Thank you for considering my suggestions.